# Continual Learning in Low-rank Orthogonal Subspaces

**Arslan Chaudhry**[1], **Naeemullah Khan**[1], **Puneet K. Dokania**[1,2], **Philip H. S. Torr**[1]

University of Oxford[1] & Five AI Ltd., UK[2]
arslan.chaudhry@eng.ox.ac.uk

## Abstract

In continual learning (CL), a learner is faced with a sequence of tasks, arriving one after the other, and the goal is to remember all the tasks once the continual learning experience is finished. The prior art in CL uses episodic memory, parameter regularization or extensible network structures to reduce interference among tasks, but in the end, all the approaches learn different tasks in a joint vector space. We believe this invariably leads to interference among different tasks. We propose to learn tasks in different (low-rank) vector subspaces that are kept orthogonal to each other in order to minimize interference. Further, to keep the gradients of different tasks coming from these subspaces orthogonal to each other, we learn isometric mappings by posing network training as an optimization problem over the Stiefel manifold. To the best of our understanding, we report, for the first time, strong results over experience-replay baseline with and without memory on standard classification benchmarks in continual learning.[1]

## 1 Introduction

In continual learning, a learner experiences a sequence of tasks with the objective to remember all or most of the observed tasks to speed up transfer of knowledge to future tasks. Learning from a diverse sequence of tasks is useful as it allows for the deployment of machine learning models that can quickly adapt to changes in the environment by leveraging past experiences. Contrary to the standard supervised learning setting, where only a single task is available, and where the learner can make several passes over the dataset of the task, the sequential arrival of multiple tasks poses unique challenges for continual learning. The chief one among which is catastrophic forgetting [McCloskey and Cohen, 1989], whereby the global update of model parameters on the present task interfere with the learned representations of past tasks. This results in the model forgetting the previously acquired knowledge.

In neural networks, to reduce the deterioration of accumulated knowledge, existing approaches modify the network training broadly in three different ways. First, *regularization-based* approaches [Kirkpatrick et al., 2016, Zenke et al., 2017, Aljundi et al., 2018, Chaudhry et al., 2018, Nguyen et al., 2018] reduce the drift in network parameters that were important for solving previous tasks. Second, *modular* approaches [Rusu et al., 2016, Lee et al., 2017] add network components as new tasks arrive. These approaches rely on the knowledge of correct module selection at test time. Third, and perhaps the strongest, *memory-based* approaches [Lopez-Paz and Ranzato, 2017, Hayes et al., 2018, Isele and Cosgun, 2018, Riemer et al., 2019], maintain a small replay buffer, called episodic memory, and mitigate catastrophic forgetting by replaying the data in the buffer along with the new task data. One common feature among all the three categories is that, in the end, all the tasks are learned in the same

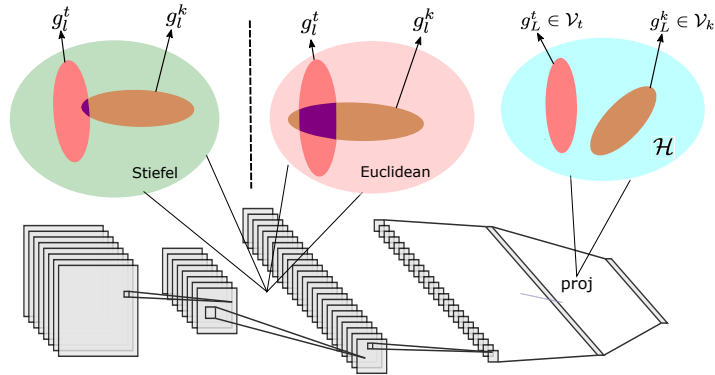

Figure 1: ORTHOG-SUBSPACE. *Each blob, with the three ellipses, represents a vector space and its subspaces at a certain layer. The projection operator in the layer L keeps the subspaces orthogonal (no overlap). The overlap in the intermediate layers is minimized when the weight matrices are learned on the Stiefel manifold.*

vector space where a vector space is associated with the output of a hidden layer of the network. We believe this restriction invariably leads to forgetting of past tasks.

In this work, we propose to learn different tasks in different vector subspaces. We require these subspaces to be orthogonal to each other in order to prevent the learning of a task from interfering catastrophically with previous tasks. More specifically, for a point in the vector space in $\mathbb{R}^m$, typically the second last layer of the network, we project each task to a low-dimensional subspace by a task-specific projection matrix $P \in \mathbb{R}^{m \times m}$, whose rank is $r$, where $r \ll m$. The projection matrices are generated offline such that for different tasks they are mutually orthogonal. This simple projection in the second last layer reduces the forgetting considerably in the shallower networks – the average accuracy increases by up to 13% and forgetting drops by up to 66% compared to the strongest experience replay baseline [Chaudhry et al., 2019b] in a three-layer network. However, in deeper networks, the backpropagation of gradients from the different projections of the second last layer do not remain orthogonal to each other in the earlier layers resulting in interference in those layers. To reduce the interference, we use the fact that a gradient on an earlier layer is a transformed version of the gradient received at the projected layer – where the transformation is linear and consists of the product of the weight matrix and the diagonal Jacobian matrix of the non-linearity of the layers in between. Reducing interference then requires this transformation to be an inner-product preserving transformation, such that, if two vectors are orthogonal at the projected layer, they remain close to orthogonal after the transformation. This is equivalent to learning orthonormal weight matrices – a well-studied problem of learning on Stiefel manifolds [Absil et al., 2009, Bonnabel, 2013]. Our approach, dubbed ORTHOG-SUBSPACE, generates two projected orthogonal vectors (gradients) – one for the current task and another for one of the previous tasks whose data is stored in a tiny replay buffer – and updates the network weights such that the weights remain on a Stiefel manifold. We visually describe our approach in Fig. 1. For the same amount of episodic memory, ORTHOG-SUBSPACE, improves upon the strong experience replay baseline by 8% in average accuracy and 50% in forgetting on deeper networks.

## 2 Background

In this section, we describe the continual learning setup followed by necessary preliminaries for our approach.

### 2.1 Continual Learning Setup

We assume a continual learner experiencing a stream of data triplets $(x_i, y_i, t_i)$ containing an input $x_i$, a target $y_i$, and a task identifier $t_i \in \mathcal{T} = \{1, \ldots, T\}$. Each input-target pair $(x_i, y_i) \in \mathcal{X} \times \mathcal{Y}_{t_i}$ is an identical and independently distributed example drawn from some unknown distribution $P_{t_i}(X, Y)$, representing the $t_i$-th learning task. We assume that the tasks are experienced in order, $t_i \leq t_j$ for all $i \leq j$, and the learner cannot store any but a few samples from $P_{t_i}$ in a tiny replay buffer $\mathcal{M}_i$. Under this setup, our goal is to estimate a predictor $f = (w \circ \Phi) : \mathcal{X} \times \mathcal{T} \to \mathcal{Y}$, composed of a

feature extractor $\Phi_\Theta : \mathcal{X} \to \mathcal{H}$, which is an $L$-layer feed-forward neural network parameterized by $\Theta = \{W_l\}_{l=1}^L$, and a classifier $w_\theta : \mathcal{H} \to \mathcal{Y}$, that minimizes the multi-task error

$$\frac{1}{T}\sum_{t=1}^T \mathbb{E}_{(x,y)\sim P_t}\left[\ell(f(x,t),y)\right],\tag{1}$$

where $\mathcal{H} \in \mathbb{R}^m$ is an inner product space, $\mathcal{Y} = \cup_{t\in\mathcal{T}}\mathcal{Y}_t$, and $\ell : \mathcal{Y} \times \mathcal{Y} \to \mathbb{R}_{\geq 0}$ is a loss function.

To further comply with the strict sequential setting, similar to prior work [Lopez-Paz and Ranzato, 2017, Riemer et al., 2019], we consider streams of data that are *experienced only once*. We only focus on classification tasks where either input or output distribution changes over time. We assume that a task descriptor, identifying the correct classification head, is given at both train and test times.

**Metrics**

Once the continual learning experience is finished, we measure two statistics to evaluate the quality of the algorithms: *average accuracy*, and *average maximum forgetting*. First, the average accuracy is defined as

$$\text{Accuracy} = \frac{1}{T}\sum_{j=1}^T a_{T,j},\tag{2}$$

where $a_{i,j}$ denotes the test accuracy on task $j$ after the model has finished experiencing task $i$. Second, the average maximum forgetting is defined as

$$\text{Forgetting} = \frac{1}{T-1}\sum_{j=1}^{T-1}\max_{l\in\{1,\dots,T-1\}}(a_{l,j} - a_{T,j}),\tag{3}$$

that is, the decrease in performance at each of the tasks between their peak accuracy and their accuracy after the continual learning experience is finished.

## 2.2 Preliminaries

Let the inner product in $\mathcal{H}$ be denoted by $\langle\cdot,\cdot\rangle$, and $v$ be an element of $\mathcal{H}$. A matrix $O \in \mathbb{R}^{m\times r}$, where $r \ll m$, parameterizes an $m \times m$ dimensional orthogonal projection matrix $P$, given by $P = O(O^\top O)^{-1}O^\top$, where $\text{rank}(P) = r$. A vector $u = Pv$, will be the projection of $v$ in a subspace $\mathcal{U} \subset \mathcal{H}$ with $\dim(\mathcal{U}) = r$. Further, if the columns of $O$ are assumed to be orthonormal, then the projection matrix is simplified to $P = OO^\top$.

**Definition 2.1** (Orthogonal Subspace). *Subspaces $\mathcal{U}$ and $\mathcal{W}$ of a vector space $\mathcal{H}$ are orthogonal if*

$$\langle u, w \rangle = 0, \quad \forall u \in \mathcal{U}, w \in \mathcal{W}.$$

**Definition 2.2** (Isometry). *A linear transformation $T : \mathcal{V} \to \mathcal{V}$ is called an isometry if it is distance preserving* i.e.
$$\|T(v) - T(w)\| = \|v - w\|, \quad \forall v, w \in \mathcal{V}.$$

A linear transformation that preserves distance, must preserve angles and vice-versa. We record this in the following theorem.

**Theorem 2.1.** *$T$ is an isometry iff it preserves inner products.*

The proof is given in Appendix A.

**Corollary 2.1.1.** *If $T_1$ and $T_2$ are two isometries then their composition $T_1 \circ T_2$ is also an isometry.*

An *orthogonal matrix* preserves inner products and therefore acts as an isometry of Euclidean space. Enforcing orthogonality[1] during network training corresponds to solving the following constrained

optimization problem:

$$\min_{\theta, \Theta = \{W_l, b_l\}_{l=1}^{L}} \ell(f(x, t), y),$$

$$\text{s.t.} \quad W_l^\top W_l = \mathbf{I}, \quad \forall l \in \{1, \cdots, L\}, \tag{4}$$

where $\mathbf{I}$ is an identity matrix of appropriate dimensions. The solution set of the above problem is a valid Riemannian manifold when the inner product is defined. It is called the Stiefel manifold, defined as $\bar{\mathcal{M}}_l = \{W_l \in \mathbb{R}^{n_l \times n_{l-1}} | W_l^\top W_l = \mathbf{I}\}$, where $n_l$ is the number of neurons in layer $l$, and it is assumed that $n_l \geq n_{l-1}$. For most of the neural network architectures this assumption holds. For a convolutional layer $W_l \in \mathbb{R}^{c_{out} \times c_{in} \times h \times w}$, we reshape it to $W_l \in \mathbb{R}^{c_{out} \times (c_{in} \cdot h \cdot w)}$.

The optimization of a differentiable cost function over a Stiefel manifold has been extensively studied in literature [Absil et al., 2009, Bonnabel, 2013]. Here, we briefly summarize the two main steps of the optimization process and refer the reader to Absil et al. [2009] for further details. For a given point $W_l$ on the Stiefel manifold, let $\mathcal{T}_{W_l}$ represent the tangent space at that point. Further, let $g_l$, a matrix, be the gradient of the loss function with respect to $W_l$. The first step of optimization projects the $g_l$ to $\mathcal{T}_{W_l}$ using a closed form $Proj_{\mathcal{T}_{W_l}}(g_l) = AW_l$, where '$A$' is a skew-symmetric matrix given by (see Appendix B for the derivation):

$$A = g_l W_l^\top - W_l g_l^\top. \tag{5}$$

Once the gradient projection in the tangent space is found, the second step is to generate a descent curve of the loss function in the manifold. The Cayley transform defines one such curve using a parameter $\tau \geq 0$, specifying the length of the curve, and a skew-symmetric matrix $U$ [Nishimori and Akaho, 2005]:

$$Y(\tau) = \left(I + \frac{\tau}{2}U\right)^{-1}\left(I - \frac{\tau}{2}U\right)W_l, \tag{6}$$

It can be seen that the curve stays on the Stiefel manifold *i.e.* $Y(\tau)^\top Y(\tau) = \mathbf{I}$ and $Y(0) = W_l$, and that its tangent vector at $\tau = 0$ is $Y'(0) = -UW_l$. By setting $U = A = g_l W_l^\top - W_l g_l^\top$, the curve will be a descent curve for the loss function. Li et al. [2020] showed that one can bypass the expensive matrix inversion in (6) by following the fixed-point iteration of the Cayley transform,

$$Y(\tau) = W_l - \frac{\tau}{2}A(W_l + Y(\tau)). \tag{7}$$

Li et al. [2020] further showed that under some mild continuity assumptions (7) converges to the closed-form (6) faster than other approximation algorithms.

## 3 Continual Learning in Orthogonal Subspaces

We now describe our continual learning approach. Consider a feature extractor in the form of a feed-forward neural network consisting of $L$ hidden layers, that takes an input $x \in \mathbb{R}^d$ and passes it through the following recursions: $h_l = \sigma(W_l h_{l-1} + b_l)$, where $\sigma(\cdot)$ is a non-linearity, $h_0 = x$, and $h_L = \phi \in \mathbb{R}^m$. The network is followed by an application-specific head (*e.g.*) a classifier in case of a classification task. The network can be thought of as a mapping, $\Phi : \mathcal{X} \to \mathcal{H}$, from one vector space ($\mathcal{X} \in \mathbb{R}^d$) to another ($\mathcal{H} \in \mathbb{R}^m$). When the network is trained for more than one tasks, a shared vector space $\mathcal{H}$ can be learned if the model has a simultaneous access to all the tasks. In continual learning, on the other hand, when tasks arrive in a sequence, learning a new task can interfere in the space where a previous task was learned. This can result in the catastrophic forgetting of the previous task if the new task is different from the previous one(s). In this work, we propose to learn tasks in *orthogonal subspaces* such that learning of a new task has minimal interference with already learned tasks.

We assume that the network is sufficiently parameterized, which often is the case with deep networks, so that all the tasks can be learned in independent subspaces. We define a family of sets $\mathcal{V}$ that partitions $\mathcal{H}$, such that, $a)$ $\mathcal{V}$ does not contain the empty set ($\emptyset \notin \mathcal{V}$), $b)$ the union of sets in $\mathcal{V}$ is equal to $\mathcal{H}$ ($\cup_{\mathcal{V}_t \in \mathcal{V}} \mathcal{V}_t = \mathcal{H}$), and $c)$ the intersection of any two distinct sets in $\mathcal{V}$ is empty ($(\forall \mathcal{V}_i, \mathcal{V}_j \in \mathcal{V}) \; i \neq j \implies \mathcal{V}_i \cap \mathcal{V}_j = \emptyset)$). A set $\mathcal{V}_t \in \mathcal{V}$ defines a subspace for task $t$. We obtain such a subspace by projecting the feature map $\phi = h_L \in \mathbb{R}^m$ into an $r$-dimensional space, where

**Algorithm 1** *Training of* ORTHOG-SUBSPACE *on sequential data* $\mathcal{D} = \{\mathcal{D}_1, \cdots, \mathcal{D}_T\}$, *with* $\Theta = \{W_l\}_{l=1}^{L}$ *initialized as orthonormalized matrices,* $\mathcal{P} = \{P_1, \cdots, P_T\}$ *orthogonal projections, learning rate '$\alpha$', $s = 2$, $q = 0.5$, $\epsilon = 10^{-8}$.*

---

1: **procedure** ORTHOG-SUBSPACE($\mathcal{D}, \mathcal{P}, \alpha, s, q, \epsilon$)
2:     $\mathcal{M} \leftarrow \{\} * T$
3:     **for** $t \in \{1, \cdots, T\}$ **do**
4:         **for** $(x_t, y_t) \sim \mathcal{D}_t$ **do**
5:             $k \sim \{1, \cdots, t-1\}$                                 $\triangleright$ Sample a past task from the replay buffer
6:             $(x_k, y_k) \sim \mathcal{M}_k$                           $\triangleright$ Sample data from the episodic memory
7:             $g^t \leftarrow \nabla_{\Theta, \theta}\left(\ell(f(x_t, y_t), P_t)\right)$                 $\triangleright$ Compute gradient on the current task
8:             $g^k \leftarrow \nabla_{\Theta, \theta}\left(\ell(f(x_k, y_k), P_k)\right)$                 $\triangleright$ Compute gradient on the past task
9:             $g \leftarrow g^t + g^k$
10:             **for** $l = \{1, \cdots, L\}$ **do**                      $\triangleright$ Layer-wise update on Stiefel manifold
11:                 $A \leftarrow g_l W_l^{\top} - W_l g_l^{\top}$
12:                 $U \leftarrow AW_l$                         $\triangleright$ Project the gradient onto the tangent space
13:                 $\tau \leftarrow \min(\alpha, 2q/(||W_l|| + \epsilon))$       $\triangleright$ Select adaptive learning rate Li et al. [2020]
14:                 $Y^0 \leftarrow W_l - \tau U$                  $\triangleright$ Iterative estimation of the Cayley Transform
15:                 **for** $i = \{1, \cdots, s\}$ **do**
16:                     $Y^i \leftarrow W_l - \frac{\tau}{2}A(W_l + Y^{i-1})$
17:                 **end for**
18:                 $W_l \leftarrow Y^s$
19:             **end for**
20:             $\theta \leftarrow \theta - \alpha \cdot g_{L+1}$                          $\triangleright$ Update the classifier head
21:             $\mathcal{M}_t \leftarrow (x_t, y_t)$                         $\triangleright$ Add the sample to a ring buffer
22:         **end for**
23:     **end for**
24:     **return** $\Theta, \theta$
25: **end procedure**

---

$r \ll m$, via a projection matrix $P_t \in \mathbb{R}^{m \times m}$ of rank $r$, *i.e.* we obtain the features for task $t$ by $\phi_t = P_t h_L$, while ensuring:

$$P_t^{\top} P_t = \mathbf{I},$$
$$P_t^{\top} P_k = \mathbf{0}, \quad \forall k \neq t. \tag{8}$$

The said projection matrix can be easily constructed by first generating a set of $m$ random orthonormal basis[1] in $\mathbb{R}^m$, then picking $r = \lfloor m/T \rfloor$ of those basis (matrix columns) to form a matrix $O_t$, and, finally, obtaining the projections as $P_t = O_t O_t^{\top}$. For different tasks these basis form a disjoint set $\mathcal{P} = \{P_1, \cdots, P_T\}$. If the total number of tasks exceeds $T$, then one can potentially dynamically resize the $m \times m$ orthogonal matrix while maintaining the required properties. For example, to make space for $2T$ tasks one can resize the original matrix to $2m \times 2m$ with zero padding, and backup the previous matrix. This would entail dynamically resizing the second last layer of the network. The set $\mathcal{P}$ can be computed offline and stored in a hash table that can be readily fetched given a task identifier. The projection only adds a single matrix multiplication in the forward pass of the network making it as efficient as standard training.

Next, lets examine the effect of the projection step on the backward pass of the backpropagation algorithm. For a task $t$, the gradient of the objective $\ell(\cdot, \cdot)$ on any intermediate layer $h_l$ can be decomposed using the chain rule as,

$$g_l^t = \frac{\partial \ell}{\partial h_l} = \left(\frac{\partial \ell}{\partial h_L}\right)\frac{\partial h_L}{\partial h_l} = \left(\frac{\partial \phi_t}{\partial h_L}\frac{\partial \ell}{\partial \phi_t}\right)\frac{\partial h_L}{\partial h_l},$$
$$= \left(P_t \frac{\partial \ell}{\partial \phi_t}\right)\prod_{k=l}^{L-1}\frac{\partial h_{k+1}}{\partial h_k} = g_L^t \prod_{k=l}^{L-1} D_{k+1} W_{k+1}, \tag{9}$$

where $D_{k+1}$ is a diagonal matrix representing the Jacobian of the pointwise nonlinearity $\sigma_{k+1}(\cdot)$. For a ReLU nonlinearity and assuming that the non-linearity remains in the linear region during the

training [Serra et al., 2017, Arora et al., 2019], we assume the Jacobian matrix to be an identity. It can be seen that for the projected layer $L$ (the second last layer), the gradients of different tasks are orthogonal by construction *i.e.* $g_L^t \perp g_L^{k \neq t}$ (*c.f.* (8)). Hence the gradient interference will be zero at the layer $L$. However, according to (9), as the gradients are backpropagated to the previous layers they start to become less and less orthogonal (*c.f.* Fig. 3). This results in interference among different tasks in earlier layers, especially when the network is relatively deep.

Let us rewrite the gradients at the intermediate layer $l$ during the training of task $t$ as a linear transformation of the gradient at the layer $L$ *i.e.* $g_l^t = T(g_L^t)$. According to (9), and assuming the Jacobian matrix of the non-linearity to be identity ($D_k = I$), this transformation is given by

$$T(u) = u \prod_{k=l}^{L-1} W_{k+1}. \tag{10}$$

As noted earlier, $g_L^t \perp g_L^{k \neq t}$ by construction, then to reduce the interference between any $g_l^t$ and $g_l^{k \neq t}$, the transformation $T(\cdot)$ in (10) needs to be such that it preserves the inner-product between $T(g_L^t)$ and $T(g_L^{k \neq t})$. In other words, $T(\cdot)$ needs to be an isometry 2.2. As discussed in Sec. 2.2, this is equivalent to ensuring that weight matrices $\{W_l\}_{l=1}^L$ are orthonormal matrices.

We learn orthonormal weights of a neural network by posing the network training as an optimization problem over a Stiefel manifold [Absil et al., 2009, Bonnabel, 2013]. More specifically, the network is initialized from random orthonormal matrices. A tiny replay buffer, storing the examples of past tasks ($k < t$), is maintained to compute the gradients $\{g_l^k\}_{l=1}^L$. The gradients on the current task $t$, $\{g_l^t\}_{l=1}^L$, are computed and weights in each layer $l$ are updated as follows: $a$) first the effective gradient $g_l = g_l^t + g_l^k$ is projected onto the tangent space at the current estimate of the weight matrix $W_l$, $b$) the iterative Cayley Transform (7) is used to retract the update to the Stiefel manifold. The projection onto the tangent space is carried out using the closed-form described in (5). The resulting algorithm keeps the network weights orthonormal throughout the continual learning experience while reducing the loss using the projected gradients. Fig. 2 shows this orthonormality reduces the inner product between the gradients of different tasks. We denote our approach as ORTHOG-SUBSPACE and provide the pseudo-code in Alg. 1.

## 4 Experiments

We now report experiments on continual learning benchmarks in classification tasks.

### 4.1 Continual Learning Benchmarks

We evaluate *average accuracy* (2) and *forgetting* (3) on four supervised classification benchmarks. **Permuted MNIST** is a variant of the MNIST dataset of handwritten digits [LeCun, 1998] where each task applies a fixed random pixel permutation to the original dataset. **Rotated MNIST** is another variant of MNIST, where each task applies a fixed random image rotation (between 0 and 180 degrees) to the original dataset. Both of the MNIST benchmark contain 23 tasks, each with 10000 samples from 10 different classes. **Split CIFAR** is a variant of the CIFAR-100 dataset [Krizhevsky and Hinton, 2009, Zenke et al., 2017], where each task contains the data pertaining 5 random classes (without replacement) out of the total 100 classes. **Split miniImageNet** is a variant of the ImageNet dataset [Russakovsky et al., 2015, Vinyals et al., 2016], containing a subset of images and classes from the original dataset. Similar to Split CIFAR, in Split miniImageNet each task contains the data from 5 random classes (without replacement) out of the total 100 classes. Both CIFAR-100 and miniImageNet contain 20 tasks, each with 250 samples from each of the 5 classes.

Similar to Chaudhry et al. [2019a], for each benchmark, the first 3 tasks are used for hyper-parameter tuning (grids are available in Appendix D). The learner can perform multiple passes over the datasets of these three initial tasks. We assume that the continual learning experience begins after these initial tasks and ignore them in the final evaluation.

### 4.2 Baselines

We compare ORTHOG-SUBSPACE against several baselines which we describe next. **Finetune** is a vanilla model trained on a data stream, without any regularization or episodic memory. **ICARL**

Table 1: *Accuracy (2) and Forgetting (3) results of continual learning experiments. When used, episodic memories contain up to one example per class per task. Last row is a multi-task oracle baseline.*

| METHOD | MEMORY | PERMUTED MNIST | | ROTATED MNIST | |
|---|---|---|---|---|---|
| | | ACCURACY | FORGETTING | ACCURACY | FORGETTING |
| FINETUNE | ✗ | 50.6 (±2.57) | 0.44 (±0.02) | 43.1 (±1.20) | 0.55 (±0.01) |
| EWC [KIRKPATRICK ET AL., 2016] | ✗ | 68.4 (±0.76) | 0.25 (±0.01) | 43.6 (±0.81) | 0.53 (±0.01) |
| VCL [NGUYEN ET AL., 2018] | ✗ | 51.8 (±1.54) | 0.44 (±0.01) | 48.2 (±0.99) | 0.50 (±0.01) |
| VCL-RANDOM [NGUYEN ET AL., 2018] | ✓ | 52.3 (±0.66) | 0.43 (±0.01) | 54.4 (±1.44) | 0.44 (±0.01) |
| AGEM [CHAUDHRY ET AL., 2019A] | ✓ | 78.3 (±0.42) | 0.15 (±0.01) | 60.5 (±1.77) | 0.36 (±0.01) |
| MER [RIEMER ET AL., 2019] | ✓ | 78.6 (±0.84) | 0.15 (±0.01) | 68.7 (±0.38) | 0.28 (±0.01) |
| ER-RING [CHAUDHRY ET AL., 2019B] | ✓ | 79.5 (±0.31) | 0.12 (±0.01) | 70.9 (±0.38) | 0.24 (±0.01) |
| **ORTHOG-SUBSPACE (OURS)** | ✗ | **86.6** (±0.91) | **0.04** (±0.01) | **80.1** (±0.95) | **0.14** (±0.01) |
| MULTITASK | | 91.3 | 0.0 | 94.3 | 0.0 |

| METHOD | MEMORY | SPLIT CIFAR | | SPLIT miniIMAGENET | |
|---|---|---|---|---|---|
| | | ACCURACY | FORGETTING | ACCURACY | FORGETTING |
| FINETUNE | ✗ | 42.6 (±2.72) | 0.27 (±0.02) | 36.1 (±1.31) | 0.24 (±0.03) |
| EWC [KIRKPATRICK ET AL., 2016] | ✗ | 43.2 (±2.77) | 0.26 (±0.02) | 34.8 (±2.34) | 0.24 (±0.04) |
| ICARL [REBUFFI ET AL., 2017] | ✓ | 46.4 (±1.21) | 0.16 (±0.01) | - | - |
| AGEM [CHAUDHRY ET AL., 2019A] | ✓ | 51.3 (±3.49) | 0.18 (±0.03) | 42.3 (±1.42) | 0.17 (±0.01) |
| MER [RIEMER ET AL., 2019] | ✓ | 49.7 (±2.97) | 0.19 (±0.03) | 45.5 (±1.49) | 0.15 (±0.01) |
| ER-RING [CHAUDHRY ET AL., 2019B] | ✓ | 59.6 (±1.19) | 0.14 (±0.01) | 49.8 (±2.92) | 0.12 (±0.01) |
| **ORTHOG-SUBSPACE (OURS)** | ✓ | **64.3** (±0.59) | **0.07** (±0.01) | **51.4** (±1.44) | **0.10** (±0.01) |
| MULTITASK | | 71.0 | 0.0 | 65.1 | 0.0 |

[Rebuffi et al., 2017] is a *memory-based* method that uses knowledge-distillation [Hinton et al., 2014] and episodic memory to reduce forgetting. **EWC** [Kirkpatrick et al., 2016] is a *regularization-based* method that uses the Fisher Information matrix to record the parameter importance. **VCL** [Nguyen et al., 2018] is another *regularization-based* method that uses variational inference to approximate the posterior distribution of the parameters which is regularized during the continual learning experience. **AGEM** [Chaudhry et al., 2019a] is a *memory-based* method similar to [Lopez-Paz and Ranzato, 2017] that uses episodic memory as an optimization constraint to reduce forgetting on previous tasks. **MER** [Riemer et al., 2019] is another *memory-based* method that uses first-order meta-learning formulation [Nichol and Schulman, 2018] to reduce forgetting on previous tasks. **ER-Ring** [Chaudhry et al., 2019b] is the strongest *memory-based* method that jointly trains new task data with that of the previous tasks. Finally, **Multitask** is an oracle baseline that has access to all data to optimize (1). It is useful to estimate an upper bound on the obtainable Accuracy (2).

## 4.3 Code, Architecture and Training Details

Except for VCL, all baselines use the same unified code base which is made publicly available. For VCL [Nguyen et al., 2018], the official implementation is used which only works on fully-connected networks. All baselines use the same neural network architectures: a fully-connected network with two hidden layers of 256 ReLU neurons in the MNIST experiments, and a standard ResNet18 [He et al., 2016] in CIFAR and ImageNet experiments. All baselines do a single-pass over the dataset of a task, except for episodic memory that can be replayed multiple times. The task identifiers are used to select the correct output head in the CIFAR and ImageNet experiments. Batch size is set to 10 across experiments and models. A tiny ring memory of 1 example per class per task is stored for the memory-based methods. For ORTHOG-SUBSPACE, episodic memory is not used for MNIST experiments, and the same amount of memory as other baselines is used for CIFAR100 and miniImageNet experiments. All experiments run for five different random seeds, each corresponding to a different dataset ordering among tasks, that are fixed across baselines. Averages and standard deviations are reported across these runs.

## 4.4 Results

Tab. 1 shows the overall results on all benchmarks. First, we observe that on relatively shallower networks (MNIST benchmarks), even without memory and preservation of orthogonality during the network training, ORTHOG-SUBSPACE outperform the strong memory-based baselines by a large

Table 2: *Systematic evaluation of Projection, Memory and Orthogonalization in* ORTHOG-SUBSPACE.

| METHOD | | | SPLIT CIFAR | | SPLIT MINIIMAGENET | |
|---|---|---|---|---|---|---|
| PROJECTION | ER | STIEFEL | ACCURACY | FORGETTING | ACCURACY | FORGETTING |
| ✓ | ✗ | ✗ | 50.3 (±2.21) | 0.21 (±0.02) | 40.1 (±2.16) | 0.20 (±0.02) |
| ✗ | ✓ | ✗ | 59.6 (±1.19) | 0.14 (±0.01) | 49.8 (±2.92) | 0.12 (±0.01) |
| ✓ | ✓ | ✗ | 61.2 (±1.84) | 0.10 (±0.01) | 49.5 (±2.21) | 0.11 (±0.01) |
| ✓ | ✓ | ✓ | 64.3 (±0.59) | 0.07 (±0.01) | 51.4 (±1.44) | 0.10 (±0.01) |

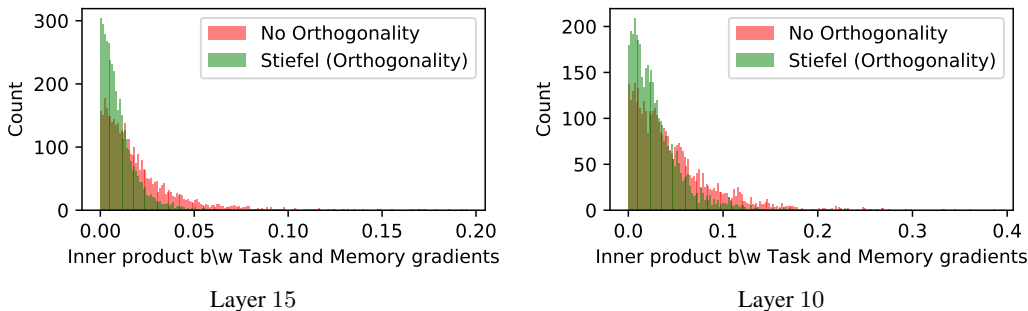

Figure 2: *Histogram of inner product of current task and memory gradients in different layers in Split CIFAR. The more left the distribution is the more orthogonal the gradients are and the less the interference is between the current and previous tasks.*

margin: +7.1% and +9.2% absolute gain in average accuracy, 66% and 42% reduction in forgetting compared to the strongest baseline (ER-Ring), on Permuted and Rotated MNIST, respectively. This shows that learning in orthogonal subspaces is an effective strategy in reducing interference among different tasks. Second, for deeper networks, when memory is used and orthogonality is preserved, ORTHOG-SUBSPACE improves upon ER-Ring considerably: 4.7% and 1.6% absolute gain in average accuracy, 50% and 16.6% reduction in forgetting, on CIFAR100 and miniImageNet, respectively. While we focus on tiny episodic memory, in Tab. 3 of Appendix C, we provide results for larger episodic memory sizes. Our conclusions on tiny memory hold, however, the gap between the performance of ER-Ring and ORTHOG-SUBSPACE gets reduced as the episodic memory size is increased. The network can sufficiently mitigate forgetting by relearning on a large episodic memory.

Tab. 2 shows a systematic evaluation of projection (8), episodic memory and orthogonalization (7) in ORTHOG-SUBSPACE. First, without memory and orthogonalization, while a simple projection yields competitive results compared to various baselines (*c.f.* Tab. 1), the performance is still a far cry from ER-Ring. However, when the memory is used along with subspace projection one can already see a better performance compared to ER-Ring. Lastly, when the orthogonality is ensured by learning on a Stiefel manifold, the model achieves the best performance both in terms of accuracy and forgetting.

Finally, Fig. 2 shows the distribution of the inner product of gradients between the current and previous tasks, stored in the episodic memory. Everything is kept the same except in one case the weight matrices are learned on the Stiefel manifold while in the other no such constraint is placed on the weights. We observe that when the weights remain on the Stiefel manifold, the distribution is more peaky around zero. This empirically validates our hypothesis that by keeping the transformation (10) isometric, the gradients of different tasks remain near orthogonal to one another in all the layers.

## 5 Related work

In continual learning [Ring, 1997], also called lifelong learning [Thrun, 1998], a learner faces a *sequence* of tasks without storing the complete datasets of these tasks. This is in contrast to *multitask learning* [Caruana, 1997], where the learner can simultaneously access data from all tasks. The objective in continual learning is to avoid catastrophic forgetting The main challenge in continual learning is to avoid catastrophic forgetting [McCloskey and Cohen, 1989, McClelland et al., 1995, Goodfellow et al., 2013] on already seen tasks so that the learner is able to learn new tasks quickly. Existing literature in continual learning can be broadly categorized into three categories.

First, *regularization approaches* reduce the drift in parameters important for past tasks [Kirkpatrick et al., 2016, Aljundi et al., 2018, Nguyen et al., 2018, Zenke et al., 2017]. For the large number of tasks, the parameter importance measures suffer from brittleness as the locality assumption embedded in the regularization-based approaches is violated [Titsias et al., 2019]. Furthermore, Chaudhry et al. [2019a] showed that these approaches can only be effective when the learner can perform multiple passes over the datasets of each task – a scenario not assumed in this work. Second, *modular approaches* use different network modules that can be extended for each new task [Fernando et al., 2017, Aljundi et al., 2017, Rosenbaum et al., 2018, Chang et al., 2018, Xu and Zhu, 2018, Ferran Alet, 2018]. By construction, modular approaches have zero forgetting, but their memory requirements increase with the number of tasks [Rusu et al., 2016, Lee et al., 2017]. Third, *memory approaches* maintain and replay a small episodic memory of data from past tasks. In some of these methods [Li and Hoiem, 2016, Rebuffi et al., 2017], examples in the episodic memory are replayed and predictions are kept invariant by means of distillation [Hinton et al., 2014]. In other approaches [Lopez-Paz and Ranzato, 2017, Chaudhry et al., 2019a, Aljundi et al., 2019] the episodic memory is used as an optimization constraint that discourages increases in loss at past tasks. Some works [Hayes et al., 2018, Riemer et al., 2019, Rolnick et al., 2018, Chaudhry et al., 2019b, 2020] have shown that directly optimizing the loss on the episodic memory, also known as experience replay, is cheaper than constraint-based approaches and improves prediction performance. Recently, Prabhu et al. [2020] showed that training at test time, using a greedily balanced collection of episodic memory, improved performance on a variety of benchmarks. Similarly, Javed and White [2019], Beaulieu et al. [2020] showed that learning transferable representations via meta-learning reduces forgetting when the model is trained on sequential tasks.

Similar in spirit to our work is OGD Farajtabar et al. [2019] where the gradients of each task are learned in the orthogonal space of all the previous tasks' gradients. However, OGD differs significantly from our work in terms of memory and compute requirements. Unlike OGD, where the memory of previous task gradients is maintained, which is equivalent to storing $n_t \times S$ dimensional matrix for each task, where $n_t$ are the number of examples in each task and $S$ is the network size, we only store $m \times r$ dimensional matrix $O_t$, where $m$ is the dimension of the feature vector ($m \ll S$) and $r$ is the rank of the subspace, and a tiny replay buffer for each task. For large network sizes, OGD is impractical to use. Furthermore, at each training step OGD subtracts the gradient projections from the space spanned by the gradients in memory, whereas we only project the feature extraction layer to a subspace and maintain orthogonality via learning on Stiefel manifolds.

Finally, learning orthonormal weight matrices has been extensively studied in literature. Orthogonal matrices are used in RNNs for avoiding exploding/ vanishing gradient problem [Arjovsky et al., 2016, Wisdom et al., 2016, Jing et al., 2017]. While the weight matrices are assumed to be square in the earlier works, works including [Huang et al., 2018] considered learning non-square orthogonal matrices (called orthonormal matrices in this work) by optimizing over the Stiefel manifolds. More recently, Li et al. [2020] proposed an iterative version of Cayley transform [Nishimori and Akaho, 2005], a key component in optimizing over Stiefel manifolds. Whereas optimizing over the Stiefel manifold ensure strict orthogonality in the weights, [Jia et al., 2019] proposed an algorithm, Singular Value Bounding (SVB), for soft orthogonality constraints. We use strict orthogonality in this work and leave the exploration of soft orthogonality for future research.

# 6   Conclusion

We presented ORTHOG-SUBSPACE, a continual learning method that learns different tasks in orthogonal subspaces. The gradients in the projected layer are kept orthogonal in earlier layers by learning isometric transformations. The isometric transformations are learned by posing the network training as an optimization problem over the Stiefel manifold. The proposed approach improved considerably over strong memory replay-based baselines in standard continual learning benchmarks of image classification.

# 7   Broader Impact

Continual learning methods like the one we propose allow machine learning models to efficiently learn on new data without requiring constant retraining on previous data. This type of learning can be useful when the model is expected to perform in multiple environments and a simultaneous retraining on all the environments is not feasible. However, one of the core assumptions in continual learning is

that model should have zero forgetting on previous data. In some scenarios, partially forgetting old data may be acceptable or even preferable, for example if older data was more biased (in any sense) than more recent data. A machine learning practitioner should be aware of this fact and use continual learning approaches only when suitable.

## Acknowledgment

This work was supported by EPSRC/MURI grant EP/N019474/1, Facebook (DeepFakes grant), Five AI UK, and the Royal Academy of Engineering under the Research Chair and Senior Research Fellowships scheme. AC is funded by the Amazon Research Award (ARA) program.

## Footnotes

[1]Code: https://github.com/arslan-chaudhry/orthog_subspace

[1]Note, an orthogonal matrix is always square. However, the matrices we consider can be non-square. In this work, the orthogonal matrix is used in the sense of $W^\top W = \mathbf{I}$.

[1]We generate a random matrix and apply the Gram–Schmidt process offline, before the continual learning experience begins.

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
