[Supplementary Material]

# Appendix

Section A provides a proof that isometry preserves angles. Section B derives the closed-form of the gradient projection on the tangent space at a point in the Stiefel manifold. Section C gives further experimental results. Section D lists the grid considered for hyper-parameters.

## A   Isometry Preserves Angles

**Theorem A.1.** *$T$ is an isometry iff it preserves inner products.*

*Proof.* Suppose $T$ is an isometry. Then for any $v, w \in V$,

$$\|T(v) - T(w)\|^2 = \|v - w\|^2$$
$$\langle T(v) - T(w), T(v) - T(w)\rangle = \langle v - w, v - w\rangle$$
$$\|T(v)\|^2 + \|T(w)\|^2 - 2\langle T(v), T(w)\rangle = \|v\|^2 + \|w\|^2 - 2\langle v, w\rangle.$$

Since $\|T(u)\| = \|u\|$ for any $u$ in $V$, all the length squared terms in the last expression above cancel out and we get

$$\langle T(v), T(w)\rangle = \langle v, w\rangle.$$

Conversely, if $T$ preserves inner products, then

$$\langle T(v - w), T(v - w)\rangle = \langle v - w, v - w\rangle,$$

which implies

$$\|T(v - w)\| = \|v - w\|,$$

and since $T$ is linear,

$$\|T(v) - T(w)\| = \|v - w\|.$$

This shows that $T$ preserves distance.   □

## B   Closed-form of Projection in Tangent Space

This section closely follows the arguments of Tagare [2011].

Let $\{X \in \mathbb{R}^{n \times p} | X^\top X = I\}$ defines a manifold in Euclidean space $\mathbb{R}^{n \times p}$, where $n > p$. This manifold is called the Stiefel manifold. Let $\mathcal{T}_X$ denotes a tangent space at X.

**Lemma B.1.** *Any $Z \in \mathcal{T}_X$ satisfies:*

$$Z^\top X + X^\top Z = 0$$

*i.e. $Z^\top X$ is a skew-symmetric $p \times p$ matrix.*

Note, that $X$ consists of $p$ orthonormal vectors in $\mathbb{R}^n$. Let $X_\perp$ be a matrix consisting of the additional $n - p$ orthonormal vectors in $\mathbb{R}^n$ *i.e.* $X_\perp$ lies in the orthogonal compliment of $X$, $X^\top X_\perp = 0$. The concatenation of $X$ and $X_\perp$, $[XX_\perp]$ is $n \times n$ orthonormal matrix. Then, any matrix $U \in \mathbb{R}^{n \times p}$ can be represented as: $U = XA + X_\perp B$, where $A$ is a $p \times p$ matrix, and $B$ is a $(n - p) \times p$ matrix.

**Lemma B.2.** *A matrix $Z = XA + X_\perp B$ belongs to the tangent space at a point on Stiefel manifold $\mathcal{T}_X$ iff $A$ is skew-symmetric.*

Let $G \in \mathbb{R}^{n \times p}$ be the gradient computed at $X$. Let the projection of the gradient on the tangent space is denoted by $\pi_{\mathcal{T}_X}(G)$.

**Lemma B.3.** *Under the canonical inner product, the projection of the gradient on the tangent space is given by $\pi_{\mathcal{T}_X}(G) = AX$, where $A = GX^\top - XG^\top$.*

*Proof.* Express $G = XG_A + X_\perp G_B$. Let $Z$ be any vector in the tangent space, expressed as $Z = XZ_A + X_\perp Z_B$, where $Z_A$ is a skew-symmetric matrix according to B.2. Therefore,

$$\begin{aligned}
\pi_{\mathcal{T}_X}(G) &= \text{tr}(G^\top Z), \\
&= \text{tr}((XG_A + X_\perp G_B)^\top(XZ_A + X_\perp Z_B)), \\
&= \text{tr}(G_A^\top Z_A + G_B^\top Z_B).
\end{aligned} \tag{11}$$

Writing $G_A$ as $G_A = \text{sym}(G_A) + \text{skew}(G_A)$, and plugging in (11) gives,

$$\pi_{\mathcal{T}_X}(G) = \text{tr}(\text{skew}(G_A)^\top Z_A + G_B^\top Z_B). \tag{12}$$

Let $U = XA + X_\perp B$ is the vector that represents the projection of $G$ on the tangent space at $X$. Then,

$$\begin{aligned}
\langle U, Z \rangle_c &= \text{tr}(U^\top(I - \frac{1}{2}XX^\top)Z), \\
&= \text{tr}((XA + X_\perp B)^\top(I - \frac{1}{2}XX^\top)(XZ_A + X_\perp Z_B)), \\
&= \text{tr}(\frac{1}{2}A^\top Z_A + B^\top Z_B)
\end{aligned} \tag{13}$$

By comparing (12) and (13), we get $A = 2\text{skew}(G_A)$ and $B = G_B$. Thus,

$$\begin{aligned}
U &= 2X\,\text{skew}(G_A) + X_\perp G_B, \\
&= X(G_A - G_A^\top) + X_\perp G_B, \quad \because \text{skew}(G_A) = \frac{1}{2}(G_A - G_A^\top) \\
&= XG_A - XG_A^\top + G - XG_A, \quad \because G = XG_A + X_\perp G_B \\
&= G - XG_A^\top, \\
&= G - XG^\top X, \quad \because G_A = X^\top G, \\
&= GX^\top X - XG^\top X, \\
&= (GX^\top - XG^\top)X
\end{aligned}$$

$\square$

## C  More Results

Table 3: *Accuracy* (2) *and Forgetting* (3) *results of continual learning experiments for larger episodic memory sizes.* 2, 3 *and* 5 *samples per class per task are stored, respectively. Top table is for Split CIFAR. Bottom table is for Split miniImageNet.*

| METHOD | ACCURACY | | | FORGETTING | | |
|---|---|---|---|---|---|---|
| | 2 | 3 | 5 | 2 | 3 | 5 |
| AGEM | 52.2 (±2.59) | 56.1 (±1.52) | 60.9 (±2.50) | 0.16 (±0.01) | 0.13 (±0.01) | 0.11 (±0.01) |
| ER-RING | 61.9 (±1.92) | 64.8 (±0.77) | 67.2 (±1.72) | 0.11 (±0.02) | 0.08 (±0.01) | 0.06 (±0.01) |
| ORTHOG-SUBSPACE | 64.7 (±0.53) | 66.8 (±0.83) | 67.3 (±0.98) | 0.07 (±0.01) | 0.05 (±0.01) | 0.05 (±0.01) |

| METHOD | ACCURACY | | | FORGETTING | | |
|---|---|---|---|---|---|---|
| | 2 | 3 | 5 | 2 | 3 | 5 |
| AGEM | 45.2 (±2.35) | 47.5 (±2.59) | 49.2 (±3.35) | 0.14 (±0.01) | 0.13 (±0.01) | 0.10 (±0.01) |
| ER-RING | 51.2 (±1.99) | 53.9 (±2.04) | 56.8 (±2.31) | 0.10 (±0.01) | 0.09 (±0.02) | 0.06 (±0.01) |
| ORTHOG-SUBSPACE | 53.4 (±1.23) | 55.6 (±0.55) | 58.2 (±1.08) | 0.07 (±0.01) | 0.06 (±0.01) | 0.05 (±0.01) |

## D  Hyper-parameter Selection

In this section, we report the hyper-parameters grid considered for experiments. The best values for different benchmarks are given in parenthesis.

- Multitask
  - learning rate: [0.003, 0.01, 0.03 (CIFAR, miniImageNet), 0.1 (MNIST perm, rot), 0.3, 1.0]
- Finetune
  - learning rate: [0.003, 0.01, 0.03 (CIFAR, miniImageNet), 0.1 (MNIST perm, rot), 0.3, 1.0]
- EWC
  - learning rate: [0.003, 0.01, 0.03 (CIFAR, miniImageNet), 0.1 (MNIST perm, rot), 0.3, 1.0]
  - regularization: [0.1, 1, 10 (MNIST perm, rot, CIFAR, miniImageNet), 100, 1000]
- AGEM
  - learning rate: [0.003, 0.01, 0.03 (CIFAR, miniImageNet), 0.1 (MNIST perm, rot), 0.3, 1.0]
- MER
  - learning rate: [0.003, 0.01, 0.03 (MNIST, CIFAR, miniImageNet), 0.1, 0.3, 1.0]
  - within batch meta-learning rate: [0.01, 0.03, 0.1 (MNIST, CIFAR, miniImageNet), 0.3, 1.0]
  - current batch learning rate multiplier: [1, 2, 5 (CIFAR, miniImageNet), 10 (MNIST)]
- ER-Ring
  - learning rate: [0.003, 0.01, 0.03 (CIFAR, miniImageNet), 0.1 (MNIST perm, rot), 0.3, 1.0]
- ORTHOG-SUBSPACE
  - learning rate: [0.003, 0.01, 0.03, 0.1 (MNIST perm, rot), 0.2 (miniImageNET), 0.4 (CIFAR), 1.0]

Figure 3: *Histogram of inner product of current task and memory gradients in all layers in Split CIFAR.*