[Reviews · NeurIPS 2020]

Review 1

Summary and Contributions: The paper proposes a novel replay-based continual learning method, named Orthg-subspace.To prevent knowledge interference among observed tasks, they adopt orthogonal task-specific parameter projections at each layer thus the update of model parameters for newly arriving tasks doesn't incur the forgetting problems to previous tasks.

Strengths: First, the proposed method shows outstanding performance compared to recent replay-based works with a few replay buffers. Their performance enhancement is based on the orthogonality across projected model parameters of observed tasks. This is concretely described and they suggest appropriate optimization techniques for solving their objective.

Weaknesses: The method is based on multi-head approach which requires task identities during training and inference. But it may not be practical for online setting while their setup is based on single-epoch training which fundamentally targets toward online scenario. Also, I have several concerns about the paper in the comment below.

Correctness: The suggested methodology and explanations look correct.

Clarity: The paper is well written and the methodology and experimental setting are well described.

Relation to Prior Work: This is one of the weakness of the paper I felt. The authors quantitatively validate their method to baselines in terms of accuracy. But, they do not provide intuitive quantitative/qualitative analysis for better understanding the uniqueness/contributions of Orthg-subspace, compared to other replay-based methods.

Reproducibility: Yes

Additional Feedback: I have several main concerns in the paper which deeply affect to the score. - The paper bypasses some expensive computation during training, I guess that they still require much larger wallclock time per training iteration. Could you give a numerical comparison of the training wallclock time with baselines? - The results for forgetting score is quite interesting that all baselines show only a marginal forgetting about less than 0.5%. But in my knowledge, for example, EWC is shown further severe forgetting (more than 10%) in many works [1,2,3]. What is the difference in the setting? - In line 5 of the algorithm, does k is randomly picked from previous tasks? Is there [1] Serra, Joan, et al. "Overcoming catastrophic forgetting with hard attention to the task." ICML 2018. [2] Lopez-Paz, David, and Marc'Aurelio Ranzato. "Gradient episodic memory for continual learning." NIPS 2017. [3] Chaudhry, Arslan, et al. "Efficient lifelong learning with a-gem." ICLR 2019. When I get feedback and solve my concerns, I'm willing to raise the score. ========== Post-rebuttal: I thoroughly read other reviews and author responses. I find the method can be performed within reasonable wall-clock time. But, I still have a concern that there is a lack of (quantitative and/or qualitative) explanations/analysis for better understanding the uniqueness/contributions of Orthg-subspace. I keep the score.


Review 2

Summary and Contributions: This paper introduces deep nets whose weight matrices are orthogonal (in the extension to the non-square matrix sense of the term). This is motivated by applications to continual learning on different tasks, and specifically the desire to embed an input's representation (and corresponding backpropagation of gradients) into a feature subspace orthogonal to the subspaces used for different tasks. For this purpose, the authors show that by a) adding a projection to a predefined subspace for each task, and b) enforcing that the weight matrices are orthogonal, feature embeddings and gradient updates does not interfere with each other (remain orthogonal) across different tasks. Experimentally, the authors evaluate the performance of their architecture across MNIST, CIFAR100 and ImageNet datasets, and provide ablation tests. *** Updated review *** I thank the authors for their response. I have updated my score to 6, to reflect the outcome of the reviewer discussion: in particular, the concerns raised regarding changing the size of T over time seem important to address; I believe that the proposed solution discussed by the authors in their response warrants empirical evaluation. However, my concerns regarding the benchmarking tasks have been alleviated.

Strengths: The theoretical motivation and analysis of this novel architecture is sound; the experimental results seem strong, sometimes improving by several standard deviations upon previous results. The topic of this paper (continual learning, specific design choices for neural nets) is entirely relevant to the NeurIPS community. In terms of significance, the authors compare to a benchmark set of problems from a previously published paper; I am not familiar enough with the literature to evaluate whether this set of benchmarks is significant (it appears to be). I am curious to know if the authors have investigated multi-task learning across a more diverse set of tasks (as it appears that the transformations applied in each task belongs to the same family of transformations: different rotation angles for Rotated MNIST, etc.).

Weaknesses: As mentioned above, the limitations of the setup for experiments, where all tasks appear to be fairly similar, might slightly reduce the results of this paper.

Correctness: The claims are correct, and the methodology (I have not checked the code) seems well-motivated and correct. Several clarifications might improve the readability of the theoretical results, and certain results (in particular regarding basic isometry properties) may not be necessary for the main paper.

Clarity: The paper is well-written. However, there are several minor grammar mistakes that I recommend be fixed, although they do not hinder understanding of the paper.

Relation to Prior Work: Although I am not particularly familiar with continual learning, the authors compare their method experimentally to a wide range of previous work, including fairly recent network architectures.

Reproducibility: Yes

Additional Feedback: *** Questions *** - For the last equality in Eq. (9), are you defining g_L^t = P_t dl/dh_L? If not, could you explain that equality in more detail? - In Figure 2, the inner products between the gradients across different tasks are much closer -- but not equal -- to 0. Is this entirely due to the fact that in practice, the ReLU activation across neurons is not entirely linear, or is there another factor explaining this behavior? - Relatedly, does Figure 2 show the absolute value of the inner products? If not, I would expect some gradients to have negative inner products. - Another assumption made in the theoretical analysis is that the layers are of decreasing size. Do the architectures you use follow that assumption? Have you thought of how you might generalize to architectures that do not verify this constraint? Although it is possible to define isometries in this space, I am curious to know if the Stiefel manifold learning can be generalized to this setting. *** Minor comments *** - I would recommend mentioning before Eq. (4) that the matrices considered are not square, and that orthogonal is here used as "W^T W = I" - l.18: "standard supervised learning *setting*"? - l.19: "poseS"? - l.79: "Let the inner product in H *be* denoted" - l. 126: where are you using the sets V_i? I didn't notice them being referenced after this paragraph. - l. 133 and elsewhere: "basEs" when using the plural of "basis" l. 147: "relatively deep"


Review 3

Summary and Contributions: The authors propose a method for tackling catastrophic forgetting in a continual learning setting. The core idea is to decouple the data for different tasks by mapping them into orthogonal subspaces through using a number of orthonormal matrices and then adapt propagation accordingly. Experiments on four different benchmark datasets are performed and the method is compared against several continual learning method to demonstrate effectiveness of the method.

Strengths: The area of continual learning is related to the NeurIPS community and recently has been focus of machine learning community. The idea of using orthogonal subspaces to tackle catastrophic forgetting in continual learning is novel and the authors have provided somewhat convincing empirical evaluation to support their claims.

Weaknesses: Despite having a novel core idea, I think this paper is not ready for publication and needs substantial improvement before publication: 1. Currently it seems that you need to know T because projection matrices P_t should be constructed before starting continual learning. This is a huge limitation because the very notion of "continual learning" implies that T is not known a priori because the learning agent supposedly is learning over unlimited time periods (i.e., we may even have T\rightarrow\infty) . Currently, learning task T+1 is going to invalidate your core idea because building an orthogonal P_{T+1} does not seem to be trivial. In my opinion, this constraint should be removed. 2. Currently it seems that you have assumed that ReLU functions are in the linear range in your derivations. But I think this is a highly slippery assumption. The very reason that ReLU is used is because of its nonlinearity. If all the ReLU functions were in the linear range, then any network will be equal to a two-layer shallow network, irrespective of the number of layers. Nonlinearity is absolutely essential to have meaningful deep networks. It may be OK to state that you estimate Jacobian matrices with identity matrices. You can even use empirical validation to check how well such a estimation is but assuming that ReLU functions are in the linear range is not a good assumption. I speculate this is why your method is not as effective when the base network is deep. 3. Section 2 is lengthier than being useful. Subsection 2.1 can be rewritten and presented in one paragraph without allocating space to trivial material, e.g., metrics. Subsection 2.2 is on quite trivial theoretical material that can be found in most linear algebra textbooks. Having that section in Appendix might be OK but its presentation in the current form is misleading. For example, Theorem 2.1 is presented as if it is a contribution by the authors but it is a preliminary theorem that is taught in undergraduate courses. The same is almost true about the rest of that section. I think the paper has almost no theoretical contribution but the presentation is such that it implies theoretical contribution and make the paper look mathematically rigorous, even if not intended. Using mathematical terms such as "isometry" or "Stiefel manifold" to describe your method does not increase your contribution. On the contrary, presenting ideas without relying on technical jargon widens outreach of your work. 4. Experimental Section: - As clear from Appendix C, out-performance of your method over memory-based methods is mainly because of using a very small memory buffer. I understand that you have used the same size to be fair, but having a bigger buffer is not a substantial limitations. I think it is more meaningful to allow for bigger buffers for the memory-based methods to the point that memory-based method outperform your method.As a result, the regimes in which your method outperforms can be understood better. Additionally methods that are based on generative experience replay (see below) do not use a memory buffer at all, yet are able to perform well under certain conditions, and in this sense are not limited. - ablative studies are missing in your results. For example, what happens if P_t matrices are not orthogonal? What is the effect of memory buffer size and other hyper-parameters? - Adding Figures which represent performance vs #task number is helpful. At the moment, you are reporting average performance but per-task performance is not visualized. You can include figures that report performance on tasks 1 to t after learning task t and use t as the x-axis. - Can you explain how do you select the samples that are stored in the buffer? Are you using the same stored samples for all methods? - How many runs do you use to computed std and average accuracy? - It is also helpful to mention which performance numbers are computed by you and if not, what reference have you used to copy the numbers. For example, I know the original EWC paper does not perform 23 permuted-MNIST tasks but currently it is not clear whether you have generated results for EWC or you are using results by third party. 6. Reference to a line of work on generative experience replay is missing in the related work section. I think incorporating these works is helpful because these works do not rely on a memory buffer for implementing experience replay: a. Shin H, Lee JK, Kim J, Kim J. Continual learning with deep generative replay, NeurIPS 2017 b . Kamra N, Gupta U, Liu Y. Deep generative dual memory network for continual learning. arXiv preprint arXiv:1710.10368. 2017 Oct 28. c. Seff A, Beatson A, Suo D, Liu H. Continual learning in generative adversarial nets. arXiv preprint arXiv:1705.08395. 2017 May 23. d. Rostami M, Kolouri S, Pilly PK. Complementary Learning for Overcoming Catastrophic Forgetting Using Experience Replay, AAAI 2019 These works use experience replay but do not use a memory buffer. I think comparing against 1-2 of these works can be helpful, too. ================================ Post-rebuttal comment: I think this work requires further demonstration to show that having a fixed T before training is not a limiting factor.

Correctness: Empirical methodology seems to be correct but can be improved. Please check the comment above.

Clarity: The paper is written well and following the text is straightforward. But I think it still can be improved, e.g., reducing section 2 and improving section 4.

Relation to Prior Work: The authors have covered prior work and explicitly stated their contribution but I think it might be better to move section 6 to right after section 1 and merge redundant material to make the paper more coherent.

Reproducibility: Yes

Additional Feedback:


Review 4

Summary and Contributions: This paper proposed a new continual learning method in low-rank orthogonal subspaces. The method learns tasks in different vector subspaces that are kept orthogonal to each other in order to minimize interference.

Strengths: Low-rank orthogonal subspaces are good for minimizing interferences among different tasks.

Weaknesses: The proposed method computes gradient on the past tasks when training current task. Consequently, they need replay buffer to store the data of past tasks. This is weekness of the method. Comparing the proposed method with other continual learning methods which does not use replay buffer, like the EWC method, is unfair. When developing continual learning algorithms, storing the data in previous tasks makes that the continual problem become easier.

Correctness: Yes

Clarity: I think so.

Relation to Prior Work: Yes.

Reproducibility: Yes

Additional Feedback: My concerens are addressed in the rebuttal.

[Author Response · NeurIPS 2020]

**CL in Low-Rank Orthogonal Subspaces** (Reviewer points are color coded **R1**, **R3**, **R4**, **R5**)

**R1: Computation during training/ Wall-clock time:** Our method requires one additional matrix multiplication in the second-last layer of the network in the forward pass and three additional matrix multiplications in the backward pass. On modern GPUs, the forward pass (inference time computation) is as efficient as standard training and backward pass adds a very small overhead to the overall training wall-clock time (seconds) which we record in the table below.

| Dataset | Finetune | ER | AGEM | Ours | Dataset | Finetune | ER | AGEM | Ours |
|---------|----------|-----|------|------|---------|----------|-----|------|------|
| MNIST | 280 | 309 | 522 | 317 | CIFAR | 529 | 850 | 1300 | 1117 |

**Forgetting Results:** Our forgetting results of EWC are compatible with that of Chaudhry et al, 2019 that the reviewer referred to (c.f. their Tab. 4 in the appendix). In fact, we used their codebase to develop our method and didn't modify the EWC routine. The only difference is that in our experiments, for fair comparisons, we initialize all the methods with orthogonal weights.

**Yes 'k' is randomly picked.**

**The practicality of the multi-head approach:** We believe that the jury is still out on what is the most practical setting for continual learning (single-/ multi-head, single-/ multiple-epochs) see `https://arxiv.org/abs/1909.08383` for a comprehensive survey. We don't claim a universal efficacy of our algorithm in all settings. We pick one setting that many recent works (especially the ones that the reviewer pointed out) adopted and provide our algorithmic contribution in that setting.

**R3: Defining $g_L$:** Yes, we are defining $g_L^t = P_t \frac{dl}{dh_L}$. The notation follows from lines 139-140 in the paper.

**Figure 2:** The reviewer made a very good observation. The inner products are not exactly zero because the weight matrices are not square and hence perfect orthogonalization is not observable. Regarding the ReLU's activation being in the linear region, see our response (Identity Jacobian) to R4.

**Decreasing size of layers:** Yes, the architectures we use follow the assumption that the layers are of decreasing size and for almost all the modern deep networks this assumption holds.

**R4: Changing number of tasks:** The way our method is presented in the paper, we do assume to know the total number of tasks 'T' beforehand but we don't consider this to be a critical limitation of our approach. One could dynamically resize the $m \times m$ orthogonal matrix to $2m \times 2m$ with zero padding, and backup the original matrix (similar to dynamic resizing of a hash table). This would entail dynamically expanding the second last layer of the network.

**Identity Jacobian** We somewhat agree with the reviewer about the non-linearity of ReLUs. We assume Jacobian to be identity and our principal motivation for this assumption comes from the work of Arora et al. (`https://arxiv.org/pdf/1901.08584.pdf`) – please refer to the text above their Eq. 7. However, we note that the authors identify the convergence to the linear update rule only for small networks. We will clarify this assumption more carefully in the paper.

**Section 2:** We don't claim *any* theoretical contributions in this work. Ours is only algorithmic contribution using well-known concepts from optimization and linear algebra. We apologize if the current presentation suggested otherwise. We will update/ rearrange the draft to avoid any pretense.

**Memory size and generative replay:** Performing well with the tiniest of memories is the focus of many recent continual learning works (`https://arxiv.org/pdf/1902.10486.pdf`, `https://arxiv.org/abs/2002.08165`, `https://arxiv.org/abs/1908.04742`) and it is the main motivation of our study. When the memory size is large, simple experience replay (multi-task training) performs the best and many recent works agree on that. Generative replay-based methods are problematic because of 1) they rely on learning a generative model in a continual setup which is as much, if not more, difficult than learning a discriminative model, 2) the memory requirement of storing a generative model is orders of magnitude higher than tiny/ small memories.

**Ablation:** Already provided in the paper – when $P_t$'s are not orthogonal (Table 2, row 1 and 3). Effect of the replay buffer size (appendix Tab. 3)

**Subset selection:** Already provided in the paper – Alg. 1, line 21, we use ring buffers storing the last 'k' examples from each class. We used a fixed seed thereby storing the same examples in the replay buffer across different methods.

**The number of runs for average and std:** Already provided in the paper – Line 208, we use 5 runs.

**Baseline numbers are computed by us:** Already provided in the paper – Line 198, we generate the numbers for *all* the baselines (except VCL) from the same codebase that we made available in the supplementary.

**R5:** Using episodic memory in a method cannot be described as its weakness. In fact, there is a class of continual learning method that relies on the replay buffer of past tasks. Our method falls in that category. Six of the eight methods that we compare against make use of replay buffers. EWC does not use episodic memory but the other six methods do. Comparing non-memory and memory-based methods is a common practice in the continual learning community. We do not think the reviewer's criticism of our work is grounded in the continual learning literature. We respectfully ask the reviewer to reconsider their evaluation of our work.

[Meta-Review · NeurIPS 2020]

The paper is proposing a continual learning method which basically learns each method in a separate orthogonal subspace. Since the updates for one task is in the null space of others, it does not negatively interfere them. The authors provide an extensive empirical results. All reviewers raised significant issues and authors answered them in their rebuttal. After the discussion phase, some issues were handled and some was still not satisfying to the reviewers. Specifically, the remaining major issues are: - R#1 finds the explanation lacking and ask for more intuitive explanation of the method. - R#4 find the fixed number of tasks (fixed T) assumption unacceptable, authors provide a possible solution but R#4 suggests it needs to be tested before acceptance. - Other reviewers are satisfied and gives accept scores. Since the scores are diverging, I carefully read the paper. Although I believe authors can definitely provide more intuition, it is not a reason for rejection. Current writing is clear enough to be accepted, and authors can improve it by the camera ready deadline. The fixed T issue is a real one; but, I personally find it rather acceptable as the proposed method is novel and definitely works well. Continual learning is still an evolving and new area, and such limitations can definitely be fixed in the future work. Moreover, authors also implement the proposed solution for dynamic T and test it. Authors shared the results and as an AC, I am satisfied. Hence, I would like to recommend acceptance. In the mean time, the following major issues need to be handled by the camera ready deadline: - Section 2 should be made significantly smaller since it includes some straightforward information which can go to appendix. The remaining space should be used to improve intuitive explanations as well as the new experimental results. - Dynamic T experiment is definitely interesting, should be completed and reported in the camera ready version.